# Epitaxial Growth and Characterization of Nanoscale Magnetic Topological Insulators: Cr-Doped (Bi_0.4_Sb_0.6_)_2_Te_3_

**DOI:** 10.3390/nano14020157

**Published:** 2024-01-11

**Authors:** Pangihutan Gultom, Chia-Chieh Hsu, Min Kai Lee, Shu Hsuan Su, Jung-Chung-Andrew Huang

**Affiliations:** 1Department of Physics, National Cheng Kung University, Tainan 701, Taiwan; pangihutangultom36@gmail.com (P.G.); jajay960430@gmail.com (C.-C.H.); 2Instrument Division, Core Facility Center, National Cheng Kung University, Tainan 701, Taiwan; anion3143@hotmail.com; 3Department of Applied Physics, National Kaohsiung University, Kaohsiung 811, Taiwan; 4Taiwan Consortium of Emergent Crystalline Materials, Ministry of Science and Technology, Taipei 10601, Taiwan

**Keywords:** topological insulator, Cr-doped (Bi_0.4_Sb_0.6_)_2_Te_3_, molecule beam epitaxy (MBE) technique, carrier density, carrier mobility

## Abstract

The exploration initiated by the discovery of the topological insulator (Bi*_x_*Sb_1−*x*_)_2_Te_3_ has extended to unlock the potential of quantum anomalous Hall effects (QAHEs), marking a revolutionary era for topological quantum devices, low-power electronics, and spintronic applications. In this study, we present the epitaxial growth of Cr-doped (Bi_0.4_Sb_0.6_)_2_Te_3_ (Cr:BST) thin films via molecular beam epitaxy, incorporating various Cr doping concentrations with varying Cr/Sb ratios (0.025, 0.05, 0.075, and 0.1). High-quality crystalline of the Cr:BST thin films deposited on a c-plane sapphire substrate has been rigorously confirmed through reflection high-energy electron diffraction (RHEED), X-ray diffraction (XRD), and high-resolution transmission electron microscopy (HRTEM) analyses. The existence of a Cr dopant has been identified with a reduction in the lattice parameter of BST from 30.53 ± 0.05 to 30.06 ± 0.04 Å confirmed by X-ray diffraction, and the valence state of Cr verified by X-ray photoemission (XPS) at binding energies of ~573.1 and ~583.5 eV. Additionally, the influence of Cr doping on lattice vibration was qualitatively examined by Raman spectroscopy, revealing a blue shift in peaks with increased Cr concentration. Surface characteristics, crucial for the functionality of topological insulators, were explored via Atomic Force Microscopy (AFM), illustrating a sevenfold reduction in surface roughness as the Cr concentration increased from 0 to 0.1. The ferromagnetic properties of Cr:BST were examined by a superconducting quantum interference device (SQUID) with a magnetic field applied in out-of-plane and in-plane directions. The Cr:BST samples exhibited a Curie temperature (T_c_) above 50 K, accompanied by increased magnetization and coercivity with increasing Cr doping levels. The introduction of the Cr dopant induces a transition from *n*-type ((Bi_0.4_Sb_0.6_)_2_Te_3_) to *p*-type (Cr:(Bi_0.4_Sb_0.6_)_2_Te_3_) carriers, demonstrating a remarkable suppression of carrier density up to one order of magnitude, concurrently enhancing carrier mobility up to a factor of 5. This pivotal outcome is poised to significantly influence the development of QAHE studies and spintronic applications.

## 1. Introduction

The study of the ternary (Bi*_x_*Sb_1−*x*_)_2_Te_3_ has attracted massive attention due to its intriguing surface states, which are anticipated to host exotic topological quantum effects [1]. This compound is formed by combining a 3D topological insulator (TI) Sb_2_Te_3_ (*p*-type) and Bi_2_Te_3_ (*n*-type) [2], where Sb atoms are partially substituted by Bi atoms. The Dirac point (DP) of Bi_2_Te_3_ is situated in the bulk valence band, with the Fermi level (E_F_) residing in the bulk conduction band, while Sb_2_Te_3_ exhibits reverse conditions. Exploiting complementary electron properties through the mixing of these two TIs is expected to be advantageous, with the Bi/Sb ratio playing a crucial role. Consequently, choosing a specific *x* value is essential for tuning the Fermi level and Dirac cone to achieve the desired properties of topological insulators [2]. Extensive investigations into the *x* value in (Bi*_x_*Sb_1−x_)_2_Te_3_ have been carried out, covering the entire composition range 0 ≤ *x* ≤ 1 [2,3,4,5,6]. It has been reported that for *x* above 0.33, (Bi*_x_*Sb_1−*x*_)_2_Te_3_ exhibits *n*-type conductivity, while *p*-type behavior is favored for values below this threshold [5,7]. Notably, when the *x* value is about 0.12, both the DP and E_F_ fall within the bulk energy gap, suggesting the existence of ideal topological insulating states [2]. Moreover, a lower scattering rate and higher mobility with surface–bulk coupling transport have been observed when the *x* value is about 0.4 [8].

Recently, the study of the TIs (Bi*_x_*Sb_1−*x*_)_2_Te_3_ has been broadened to magnetic topological insulators, where the quantum anomalous Hall effect (QAHE) is predicted to appear [9,10,11]. Magnetic TIs are also expected to bring revolutionary developments in topological magnetoelectric and topological quantum computation [12]. It is known that the bulk band gapped spin–split surface states are protected by the time-reversal symmetry (TRS) in TIs. Manipulation of TIs through the doping of magnetic impurities, such as chromium (Cr) and vanadium (V), can break the TRS due to the appearance of magnetic moment, and can thus open a band gap at the Dirac point and active the QAHE without the need for an external magnetic field [1,5,13,14,15,16,17]. In such scenarios, the initially protected surface state transforms into a trivial insulator (*C* = 0), devoid of chiral edge channels. Therefore, a topological quantum phase transition between the QAH (*C* = 1) and trivial-insulator (*C* = 0) phases is naturally anticipated to occur by varying the exchange interaction energy via magnetic impurities. The ability to induce such transitions offers a pathway for manipulating and controlling the unique electronic states in these materials. While the observation of the QAH effect in the Cr-doped (Sb, Bi)_2_Te_3_ system has been demonstrated, practical application is currently restricted to extremely low temperatures (<100 mK) [1,16]. To realize the full potential of magnetic TIs in practical applications, there is a pressing need for the development of ferromagnetic TIs with much higher T_c_. Moreover, the introduction of a Cr dopant plays an essential role in effectively suppressing the carrier density and improving the carrier mobility, thereby enabling the pronounced observation of substantial anomalous Hall effects (AHEs) [18]. Several studies have been conducted in Cr-doped (Cr*_y_*Bi*_x_*Sb_1−*x*−*y*_)_2_Te_3_ to determine the optimum Bi/Sb ratio and Cr dopant concentration [18,19,20,21]. However, the intricate nature of the components involved has posed challenges regarding the determination of the optimum *x*/*y* ratio. Several techniques have been employed to grow Cr-doped (Bi_x_Sb_1−*x*_)_2_Te_3_, such as vacuum thermal evaporation [22,23], the Bridgman method [24], and molecule beam epitaxy (MBE) [22,25]. Among these techniques, MBE stands out as a particularly favorable method for growing Cr-doped (Bi*_x_*Sb_1−*x*_)_2_Te_3_, offering precise control over stoichiometry, crystal quality, and surface morphology [26]. This investigation is specifically geared toward identifying the optimal temperature and Cr concentration to modulate the charge carriers, achieving a low bulk charge concentration and high carrier mobility and introducing ferromagnetism in Cr-doped (Bi_0.4_Sb_0.6_)_2_Te_3_ grown by the MBE technique. The results of this work are essential for the preparation and control of the magnetic and electrical properties of magnetic topological insulators. The detailed insights gained from this research not only contribute to advancing our understanding of the fundamental physics underlying these materials but also pave the way for the design and optimization of magnetic topological insulator-based devices with enhanced functionalities.

## 2. Experimental

The Cr-doped (Bi_0.4_Sb_0.6_)_2_Te_3_ thin films were fabricated using the MBE system (AdNaNo Corp., New Taipei City, Taiwan, model MBE-9) on sapphire substrates (0001) with dimensions of 1 × 1 cm^2^. Highly pure Cr (99.9%), Sb (99.9999%), Bi (99.9999%), and Te (99.9999%) were co-evaporated from Knudsen effusion cells. The cell temperature was precisely adjusted prior to growth to achieve the desired flux, which was calibrated with a beam flux monitor (BFM). The substrate underwent the same cleaning process as described in our prior publication [27,28,29]. After cleaning, the sapphire substrate was inserted into the MBE chamber and then preheated at 1010 °C under 10^−9^ Torr for 70 min. Subsequently, the growth of the Cr:BST process was conducted at various temperatures of about 200, 300, 400, and 500 °C, followed by natural cooling to room temperature. This work aims to identify the optimal epitaxial growth temperature of Cr:BST samples and investigate their structural, electrical, and magnetic properties. Throughout the growth process, Sb flux was maintained at 3 × 10^−8^ Torr, while Te flux was excessively supplied with a Te/Sb ratio at 17. The Bi/Sb ratio was fixed at 0.4/0.6. Additionally, various Cr doping concentrations were regulated with the Cr/Sb ratio range of 0 to 0.3, while maintaining the growth temperature at about 300 °C. The amorphous Te layers (1~2 nm) were deposited in situ after the growth of the Cr:BST films to prevent environmental contamination. RHEED was used in situ to monitor the film quality. Structural characterization was performed utilizing XRD with a Burker D8 Discover instrument (Billerica, MA, USA) equipped with a general area detector diffraction system (GADDS). Cu K-α radiation with a wavelength of 1.54184 Å was employed for the XRD analysis. Additionally, HRTEM was conducted using a selected-area electron diffractometer at an accelerating voltage of 200 kV. The surface morphology was characterized by an AFM. The effect of the Cr doping on the lattice vibration of Cr:BST was qualitatively studied using Raman spectroscopy. The Raman spectra were recorded using a micro-Raman spectrometer (Horiba-Jobin Yvon LabRam-HR, Villeneuve d’Ascq, France) equipped with a grating of 1800/mm. The excitation source employed for the Raman analysis had a wavelength of 532 nm, and all measurements were conducted at room temperature. The XPS measurements were performed at a synchrotron source with beamline TLS 24A1 of Taiwan Light Source in the National Synchrotron Radiation Research Center (NSRRC), Hsinchu, Taiwan. Various excitation energies were applied during the measurements. To ensure accuracy, the onset of photoemission from a gold foil attached to the sample holder was used to calibrate the binding energy. The magnetic property of Cr:BST was examined by a SQUID (Quantum Design MPMS SQUID VSM system, San Diego, CA, USA). An external magnetic field was applied to the surface of the epitaxial layer in a direction either in plane or out of plane. For the study of electrical transport, the Cr:BST films were patterned into a Hall bar geometry using photolithography. The Hall bar size was selected at 100 μm × 50 μm, and the entire process was conducted under low pressure of 10^−3^ Torr. This approach facilitated the measurement of longitudinal resistivity (*ρ_xx_*) and Hall resistivity (*ρ_yx_*) using a closed-cycle refrigeration system and a physical property measurement system (PPMS).

## 3. Results and Discussion

### 3.1. Cr-Doped (Bi_0.4_Sb_0.6_)_2_Te_3_ with the Cr/Sb Flux Ratio and Growth Temperature

XRD analysis was conducted to investigate the influence of temperature and Cr doping concentrations on the structure of Cr:BST films. First, we studied the growth temperature effect under a fixed Cr/Sb ratio of 0.1. A growth temperature of 200 °C is insufficient to promote the epitaxial growth of Cr:BST, as illustrated by the XRD in Figure 1a. For a temperature of 300 °C, the appearance of the diffraction lattice planes at (003), (006), (009), (0012), (0015), (0018), (0021), and (0024) confirms the ordered rhombohedral structure of the films, consistent with previous studies [22,25]. The ordered rhombohedral structure is slightly weakened as the growth temperature increases to 400 °C. When the growth temperature is further raised to 500 °C, the (00l) peaks disappear, and (112) and (224) peaks of Cr_2_Te_3_ emerge [30,31]. This indicates that the elevated temperature is sufficient to significantly reduce the condensation of Bi and Sb elements on the substrate, facilitating the formation of Cr_2_Te_3_. At the optimal growth temperature of 300 °C, the impact of Cr doping concentration on the structure of Cr:BST was studied in Figure 1b. The XRD spectra display prominent characteristic (00l) signals, with all the peaks normalized to the (0015) peaks for comparison. When the Cr/Sb ratio is in the range from 0 to 0.1, the structure of Cr:BST films reflects the ordered rhombohedral structure. Notably, the (0015), (0018), and (0021) peaks exhibit a slight shift to higher angles with an increasing Cr/Sb ratio. This phenomenon aligns with the Sherrer equation, where the shift of the XRD peaks is in proportion to the relation with the lattice parameter of the material. Therefore, the shift of the XRD peaks can be attributed to the incorporation of Cr, wherein the substitution of Bi/Sb by Cr leads to a reduction in the crystal lattice parameter. This reduction arises from the smaller ionic radius of Cr compared to Sb and Bi [32,33], as confirmed by the reduction in the *c*-axis lattice parameters of (0015) from 30.53 ± 0.05 to 30.06 ± 0.04 Å with an elevated Cr/Sb ratio, as shown in Figure 1c [34,35]. Additionally, the intensity of the signal peak (003) strengthens with the increase in Cr concentration, likely corresponding to an enhancement of surface smoothness. However, when the samples are doped with a Cr/Sb ratio above 0.2, Cr_2_Te_3_ signals appear, and the characteristic (00l) peaks of BST vanish (Appendix A). 

Figure 1d shows the HRTEM image along the *c*-axis direction, clearly revealing the characteristic quintuple-layer (QL) structure with van der Waals (vdW) stacking in a Cr:BST epitaxial film. The film thickness was identified as 25 QL. The effect of the various Cr doping concentrations on lattice vibration of the Cr:BST was qualitatively studied using Raman spectroscopy with an excitation wavelength of 532 nm at a frequency range of 100–200 cm^−1^. In Figure 1e, the BST sample exhibits three primary Raman active modes at 126.7, 141.2, and 163.5 cm^−1^, corresponding to three modes, $E_{g}^{2}$, A_1u_, and $A_{1g}^{2}$, respectively [36,37]. The A_1u_ and $A_{1g}^{2}$ modes are sensitive to the out-of-plane Bi/Sb-Te bond vibrations, while $E_{g}^{2}$ mode is sensitive to the in-plane Bi/Sb-Te bond vibration [36,38,39]. Moreover, $E_{g}^{2}$ modes involve in-plane bending, where the bonding of each atom is stronger (covalent bonds) than out-of-plane (vdW forces) [39]. It is noted that $E_{g}^{2}$ mode slightly shifts toward higher wavenumbers as the Cr/Sb ratio increases [36], while A_1u_ mode remains constant (Appendix A).

The observed blueshift in $E_{g}^{2}$ modes could be attributed to structural distortion due to the substitution of Sb or Bi by Cr atoms [40]. First-principle calculations further support this by indicating that the formation energies of Cr favor the substitution of Bi sites in BST [19]. As the Cr/Sb ratio increased, the intensity of the three modes was notably attenuated, indicating a deterioration in the bulk structural quality with an elevated Cr concentration. Additionally, to gain insights into the chemical environment of Cr within the sample, we conducted XPS measurements, focusing on the Cr core level. Unfortunately, the XPS peaks of Cr 2p_3/2_ and 2p_1/2_ orbital overlaps with the Te 3d_5/2_ and 3d_1/2_ orbitals, and the peak of the Cr 3p orbital overlaps with the Te 4d_3/2_ and 4d_5/2_ orbitals. Aside from that, the XPS signal intensities of the Cr 2s and 3s orbitals and the Te 3p_1/2_ and 3p_3/2_ orbitals are very weak. Therefore, we chose the peaks of Cr 2p and the Te 3d orbitals, which show the strongest XPS signal that can be used to confirm the chemical bond states of Cr. Figure 1f shows the deconvolution of the Te 3d core levels for the Cr/Sb = 0.1 sample. The binding energy of Te 3d_5/2_ and 3d_1/2_ core levels (blue peaks) are determined to be 572.60 eV and 582.5 eV, respectively. The Te percentage is much larger than Cr, resulting in a weaker Cr signal intensity. The discernment of the Cr signal necessitates deconvolution peaks using a multi-peak Lorentz fitting with the uncertainty shown in Appendix A. From the fitting results, the binding energy of Cr 2p_3/2_ and 2p_1/2_ core levels (green peaks) are located at 573.1 eV and 583.5 eV, respectively. Therefore, the XPS result confirms the valence states of Cr in Cr:BST samples.

The surface morphology of Cr-doped (Bi_0.4_Sb_0.6_)_2_Te_3_, grown at 300 °C with varying Cr/Sb ratios from 0 to 0.1, was quantitatively examined by AFM measurements. Figure 2a shows the AFM image of the grown BST film, indicating a reasonably flat surface characterized by distinct terraces and steps, reflecting the hexagonal crystal structure of BST grown along (0001), which is consistent with the aforementioned XRD results. With the increasing Cr/Sb ratio, the surface morphology reveals misoriented domains with ambiguous shapes, as shown in Figure 2b–e. We have also examined the surface morphology of Cr:BST grown with Cr/Sb ratios of 0.2 and 0.3, as shown in Appendix A. Additionally, the RHEED patterns consistently show sharp 1 × 1 diffraction streaks across various Cr/Sb ratios, indicating high crystalline order and relatively smooth surface morphologies (insets of Figure 2a–e). The surface roughness of Cr:BST films decreased from 2.8 to 0.4 nm as the Cr/Sb ratio increased from 0 to 0.1. These structural features, confirmed by AFM, HRTEM, and XRD, reveal the high quality of the Cr:BST films grown by the delicate MBE method.

### 3.2. Magnetic Properties

A SQUID magnetometer was employed to investigate the magnetic properties of Cr-doped (Bi_0.4_Sb_0.6_)_2_Te_3_ films with varying Cr/Sb ratios from 0 to 0.1. The diamagnetic signal due to the substrate was subtracted by fitting to the high-field data and subsequently removing the diamagnetic background (Appendix A). 

Figure 3a–c present the magnetization curves of Cr-doped (Bi_0.4_Sb_0.6_)_2_Te_3_ samples with different Cr/Sb ratios obtained at temperatures of 5, 20, and 50 K, with the magnetic field applied along the out-of-plane direction (H//c) of the samples. The out-of-plane M-H hysteresis loops reveal ferromagnetic behavior along the easy axis. Note that the out-of-plane saturation magnetization and coercivity increase with an increasing Cr/Sb ratio (Table 1 and Table 2). Evidently, the T_c_ of the Cr:BST sample with a Cr/Sb ratio of ~0.025 falls below 50 K. In contrast, the Cr:BST samples with a higher Cr/Sb ratio over 0.05 exhibit T_c_ above 50 K. Additionally, the M-H curve for the 0.05 sample shows non-square loop behavior, which might be attributed to the substrate effect [28].

The magnetic effect of the Cr doping in BST films can be elucidated through two primary mechanisms: the van Vleck mechanism and Ruderman–Kittel–Kasuya–Yosida (RKKY) coupling [19,41,42]. In the case of the van Vleck mechanism, the significant spin susceptibility of valence electrons in band-inverted TIs materials enables direct coupling of magnetic ions through these localized valence electrons, bypassing the need for itinerant electrons. This leads to what is termed “bulk ferromagnetism”, which is not dependent on carrier density [1]. On the other hand, the RKKY interaction involves coupling between neighboring magnetic ions via the mediation of conduction carriers, known as carrier-mediated RKKY interaction [41,42]. Moreover, the RKKY mechanism is particularly evident in bulk *p*-type Cr-doped BST samples [43,44]. In this case, the T_c_ is found to be proportional to $n_{p}$^1/3^, where $n_{p}$ is the hole carrier density. The ferromagnetism in this scenario is strongly related to the hybridization between the transition-metal d states and the anion p states [44]. Notably, higher Cr doping concentrations lead to elevated T_c_ values because of the shorter distances among the Cr atoms and the more robust interactions between them. The magnetization curves of Cr-doped (Bi_0.4_Sb_0.6_)_2_Te_3_ film (Cr/Sb ratio~0.1) with a magnetic field along the in-plane direction at 5, 20, and 50 K were illustrated in Figure 3d. The in-plane M-H loops flat curves, resembling a hard-axis configuration. The results suggested the Cr:BST films unveiled a perpendicular magnetic anisotropy with the easy magnetization axis along the c-axis, holding significance for the development of ferromagnetic topological insulators in the context of QAH studies and spintronic applications.

### 3.3. Transport Properties

The influence of various Cr/Sb ratios is also evident in the transport properties. The electrical properties of Cr:BST samples were studied by analyzing the longitudinal resistance R*_xx_* and Hall resistance R*_xy_* at temperatures of 16 K with a magnetic field up to 20,000 Oe. Figure 4a summarizes the evolution of sheet resistance (R_□_) and carrier mobility with various Cr/Sb ratios. The R_□_ of the Cr:BST sample shows a maximum at a Cr/Sb ratio of 0.025, subsequently decreasing as the Cr/Sb ratio further increases to 0.1, as depicted on the left side in Figure 4a. Additionally, the carrier mobility of Cr:BST also increases until a Cr/Sb ratio of 0.025 reaches its maximum and then decreases as the Cr/Sb ratio further rises to 0.1, as shown on the right side in Figure 4a. Notably, the carrier mobility of Cr:BST with a Cr/Sb ratio of 0.025 is about five times larger than the BST sample. Figure 4b displays the carrier concentration of Cr:BST samples as a function of the Cr/Sb ratio. In the absence of Cr doping, the (Bi_0.4_Sb_0.6_)_2_Te_3_ sample exhibits *n*-type conductivity, which is consistent with previous results reported for (Bi*_x_*Sb_1−_*_x_*)_2_Te_3_ [2]. The carrier density |*n*2*_D_*| of Cr:BST films reaches its minimum at a Cr/Sb ratio of 0.025 with |*n*2*_D_*|= 2.3 × 10^12^ cm^−2^ and increases on both sides. The incorporation of Cr adjusts the charge carrier in Cr:BST samples, transitioning from *n*-type to *p*-type, and the hole concentration slightly increases with higher Cr doping levels. The results suggest that the BST composition (Bi_0.4_Sb_0.6_)_2_Te_3_ is close to the desired value, allowing the Cr dopant to modulate the charge carrier type with a low bulk carrier concentration and high carrier mobility, inducing ferromagnetism in BST topological insulators. The ferromagnetic ordering in Cr:BST films was further confirmed by magneto-transport measurements. For clarity, we focus on the sample with a Cr/Sb ratio of 0.025, showing low bulk carrier concentration and high carrier mobility. Figure 4c,d display the Hall resistivity ($\rho_{xy}$) and longitudinal resistivity ($\rho_{xx}$) measured at 5 K in a magnetic field up to 6000 Oe applied perpendicular to the films. The $\rho_{xx}$ shown in Figure 4c exhibits a butterfly-shaped curve and two double-split sharp peaks located at the coercive field H_C_ that decreased rapidly on both sides, which is consistent with previous results for Cr-doped BST films [22]. 

Figure 4d displays clear hysteresis loops of anomalous Hall resistivity, implying the well-defined perpendicular ferromagnetism in the Cr:BST. However, the $\rho_{xy}$ only reaches 0.02 (*e*^2^/*h*), indicating that it is not ideal yet for achieving the QAHE. While the full quantization of the QAHE signal remains unachieved, we anticipate that further refinement of the composition of (Bi*_x_*Sb_1−*x*_)_2_Te_3_ and Cr doping levels, along with the application gate voltage on the Hall effect device, will unveil and optimize the QAHE in this system.

## 4. Conclusions

In summary, we have presented a synthesis and characterizations of Cr-doped (Bi_0.4_Sb_0.6_)_2_Te_3_ (Cr:BST) films prepared using MBE. By carefully controlling the epitaxy growth conditions, we have demonstrated the preparation of high-quality, single-crystalline Cr:BST samples, as confirmed by XRD. The AFM and HRTEM analyses further confirmed a flat surface morphology covering a large area, with evidence of van der Waals stacking. Raman spectroscopy provided insight into the impact of Cr doping on the structure, indicating a discernible blue shift in signals due to the substitution of Sb or Bi by Cr. The existence of Cr states in Cr:BST samples was unequivocally confirmed through XPS results. The magnetic behavior of Cr:BST films exhibited perpendicular magnetic anisotropy with T_c_ above 50 K. Further, the evolution of sheet resistance and carrier mobility reached peak maximum for the Cr:BST sample with a Cr/Sb ratio of 0.025, aligning with the electronic behavior displaying a transition from *n*-type to *p*-type carriers. Hall and longitudinal resistance measurements confirmed that the Cr: BST sample with a Cr/Sb ratio of 0.025 exhibited an AHE effect at 5 K. These findings collectively underscore the intricate interplay between Cr doping levels and the resulting structural, magnetic, and electronic properties of Cr-doped (Bi_0.4_Sb_0.6_)_2_Te_3_ films. Although the complete quantization of the quantum anomalous Hall effect signal has not yet been accomplished, we foresee that through additional fine-tuning of the composition of (Bi*_x_*Sb_1−*x*_)_2_Te_3_ and the levels of Cr doping, coupled with the application of gate voltage on the Hall effect device, we can reveal and optimize the QAHE in this system. This study provides valuable insights for applications in emerging technologies, such as the quantum anomalous Hall effect and spintronic applications, based on magnetic topological insulators.

## Figures and Tables

**Figure 1 nanomaterials-14-00157-f001:**
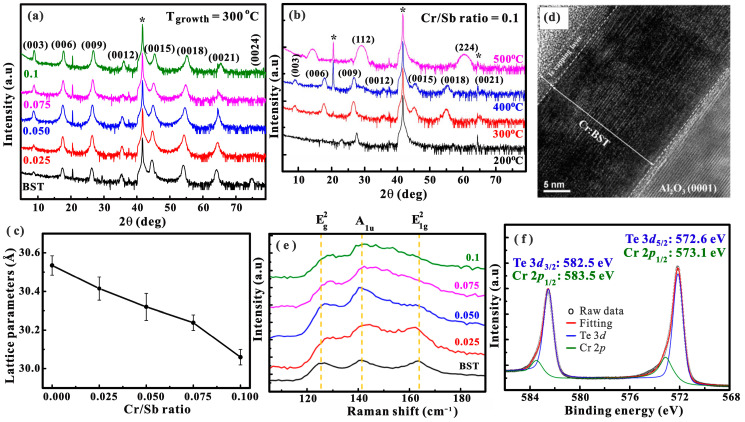
XRD diagrams for (**a**) a fixed Cr/Sb ratio of 0.1 under varied growth temperatures (indicated on the right-hand axis) and (**b**) a fixed growth temperature at 300 °C with varied Cr/Sb ratios (indicated on the right-hand axis). The signals are marked as asterisks for the c-plane Al_2_O_3_ substrate indicated in the figure. (**c**) *c*-axis lattice parameter extracted from (0015) peak as a function of the Cr/Sb ratio. (**d**) TEM cross-sectional view of the film with a Cr/Sb ratio of 0.1 and a growth temperature of 300 °C. (**e**) Raman spectra of films with varied Cr/Sb ratios, with dashed vertical lines highlighting the three Raman active modes, $E_{g}^{2}$, A_1*u*_, and $A_{1g}^{2}$. (**f**) XPS spectra of Cr 2p and Te 3d orbitals of the Cr/Sb ratio of the 0.1 sample. The black dots are the XPS spectra, and the blue and green curves are the Lorentz fitting curves corresponding to the Cr 2p and Te 3d orbitals, while the red curves are the total fitting curves. Note: * sign: c-plane Al_2_O_3_ substrate signals.

**Figure 2 nanomaterials-14-00157-f002:**
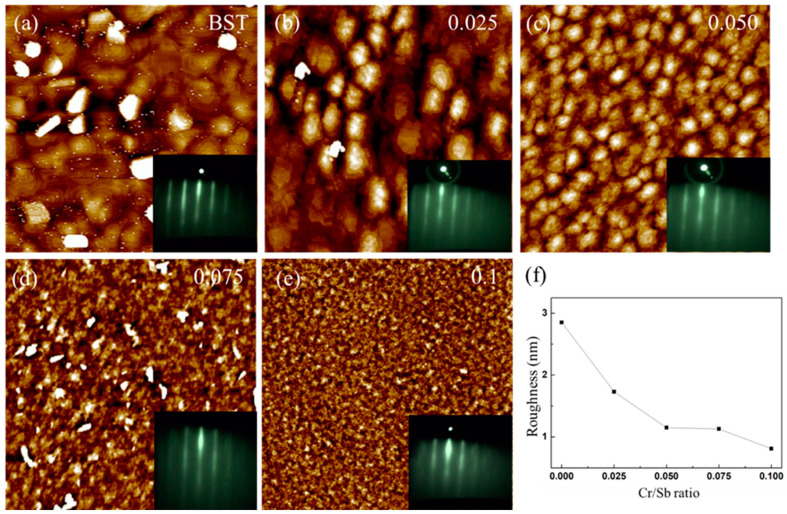
(**a**–**e**) Dependence of film morphology on Cr:BST samples with varied Cr/Sb ratios from an AFM (all images have a scan size of 2 μm × 2 μm). Inset corresponds to RHEED patterns. (**f**) Roughness for varied Cr/Sb ratios.

**Figure 3 nanomaterials-14-00157-f003:**
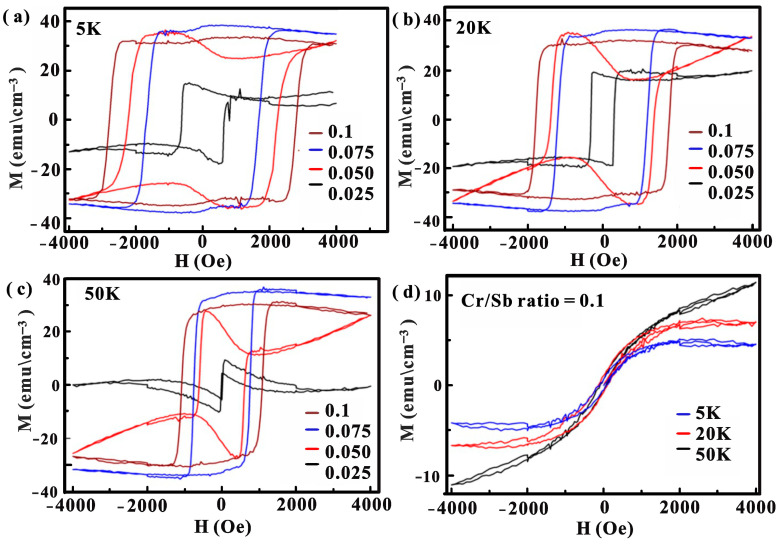
(**a**–**c**) Hysteresis loops of the Cr:BST samples with various Cr/Sb ratios in a magnetic field along an out-of-plane direction at different temperatures of 5, 20, and 50 K, respectively. (**d**) Hysteresis loops of a Cr/Sb ratio of a 0.1 sample in a magnetic field along the in-plane direction at different temperatures.

**Figure 4 nanomaterials-14-00157-f004:**
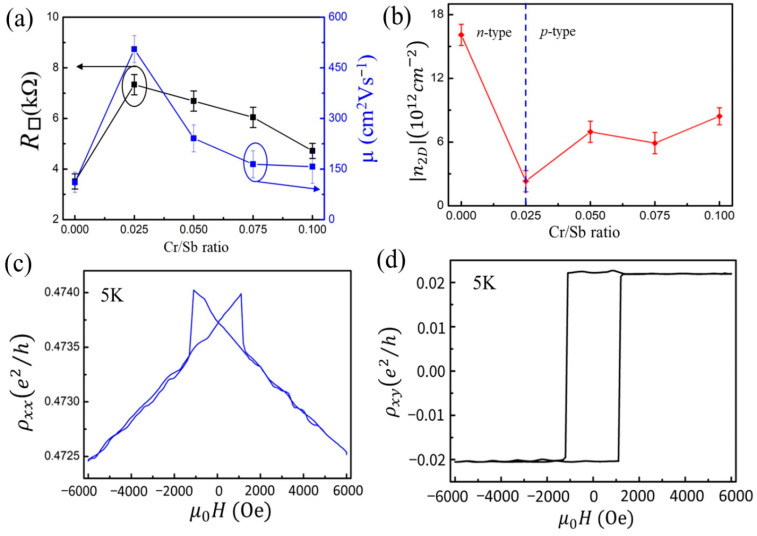
(**a**) The sheet resistance (R_□_) (left ordinate axis) and mobility (left ordinate axis) of Cr:BST samples with varied Cr/Sb ratios. (**b**) The absolute value of the sheet electron concentration *n*_2*D*_ of Cr:BST with varied Cr/Sb ratios. (**c**) Magnetic field-dependent longitudinal resistance curves at 5 K of the Cr:BST sample with a Cr/Sb ratio of 0.025. (**d**) A rectangular-shaped hysteresis loop in magnetic field-dependent Hall resistance curves at 5 K of the Cr:BST sample with a Cr/Sb ratio = 0.025.

**Table 1 nanomaterials-14-00157-t001:** The temperature-dependent saturation magnetization of Cr-doped (Bi_0.4_Sb_0.6_)_2_Te_3_ at different Cr/Sb ratios.

Temperatures (K)	Cr/Sb Ratio			
	0.025	0.05	0.075	0.1
5	17.5	36.6	38.5	33.9
20	18.7	36.0	37.7	32.5
50	-	27.9	34.7	30.9

Notes: (unit of saturation magnetization: emu/cm^−3^).

**Table 2 nanomaterials-14-00157-t002:** The temperature-dependent coercivity of Cr-doped (Bi_0.4_Sb_0.6_)_2_Te_3_ with different Cr/Sb ratios.

Temperatures (K)	Cr/Sb Ratio			
	0.025	0.05	0.075	0.1
5	630	2230	1680	2800
20	300	1370	1190	1780
50	-	610	750	1080

Notes: (unit of coercivity: Oe).

## Data Availability

All data included in this study are available upon request by contacting the corresponding author.

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
