# Peer review of "Epitaxial Growth and Characterization of Nanoscale Magnetic Topological Insulators: Cr-Doped (Bi0.4Sb0.6)2Te3"

_nanomaterials, 2024, doi:10.3390/nano14020157_

Round 1
Reviewer 1 Report
Comments and Suggestions for Authors
The work is impressive, and I appreciate the hard work put in for the manuscript. The material characterization is detailed and thorough. I would appreciate it if the authors clarified some points mentioned below and made appropriate changes to the manuscript.
1. The transport properties show the best/optimum properties at x=0.025, whereas the magnetic properties in Curie temperature are highest at x=0.075. Would the authors comment on which doping is preferred or what the overarching goal of the study is? Is it demonstrating high Tc can or best samples for transport and observing QAHE? I am asking because it is commented in line 70 “the practical application is currently restricted to extremely low temperatures (< 100 mK)”, presumably low Tc being a reason. But higher Tc does not necessarily make it better for QAHE, as your data shows. Moreover, thermally activated behavior will occur at higher temperatures, even in a “perfect” high-Tc sample. So, the quantum hall plateaus will never be quantized to a high accuracy at higher temps. Therefore, I am a bit skeptical about the search for high-Tc materials for QAHE if, no matter what the samples are, the measurements must always be done at extremely low temperatures. This knowledge will be essential to motivate a broader audience.
2. The mobility in your samples seems a bit low compared to Chang et al. (Science 2010, x ~ 0.09), where they observed QAHE. Rho_xx data, apart from Rho_xy, also seems to indicate significant disorder in your best sample (x=0.025).
What could be the possible reasons, and could you outline the strategies (other than gate voltage) in your growth process to improve carrier mobility? It is commented in Line 311 “we anticipate that further refinement of the composition of (BixSb1-x)2Te3 and Cr doping levels”. Some more details will be useful here since I am not sure simply refining the doping levels will be enough.
3. No data is shown at lower than 5 K or higher than 0.6T, Is there any particular reason why lower temp/higher field data is not reported?
4. The magnetization behavior shows non-standard/non-square out-of-plane hysteresis loops for some compositions (x=0.05, for example), indicating non-trivial exchange coupling in the system. Please comment on some possible reasons.
Author Response
Please read the attachment.

Reviewer 2 Report
Comments and Suggestions for Authors
The authors presented an interesting work in magnetic topological insulator study and aimed on achieving quantum anomalous Hall effect in higher temperatures. The fine manipulations of chemical potential and magnetic ordering in high quality thin film are essential for QAHE study. Therefore, the present work contributes to significant progress in the magnetic topological insulator research area, and can be published in Nanomaterials after addressing the following comments:
1. Line 53, 54, topological insulating state is intrinsic property to a material, which not depends on where DP and Fermi sit. I guess the authors are claiming the band structures, with DP sitting in bulk gap, and Fermi close to DP, are ideal to study intrinsic properties of TI.
2. Line 56, Hall.
3. Line 59, the “spin-split surface states” is a very unusual expression.
4. I don’t understand the topological states in Line 63-66. Chern insulators?
5. Line 71-72, high Tc is not enough.
6. Line 86, why Bi0.4?
7. Line 157, what is the error bar of lattice constant in Fig 1c?
8. Line 163, van der Waals.
9. Fig. 1b, the intensity of (003) peak changes with Cr/Sn ratio. The normalization to (0015) peak intensity is very usual, which should be normalized to substrate intensities.
10. Legends in Fig. 3 are too small to read.
11. Incorrect background subtraction was used in Fig. 3, please work on this again, and show the detail of original data, subtraction methods in the supplementary materials. Moreover, MT curves are also very important to show the magnetic phase transitions.
12. Line 384, why 16 K?
13. Please add full sets of transport data in the manuscript.
Comments on the Quality of English Language
none.
Author Response
Please read the attachment.

Reviewer 3 Report
Comments and Suggestions for Authors
This manuscript reports on exhaustive characterization studies of the structural, magnetic and electronic properties of Cr-doped BST thin films, where they systematically vary the Cr/Sb ratio. The authors describe the thin film synthesis using MBE and then report on structural characterization using electron and x-ray diffraction as well as AFM measurements and Raman spectroscopy. They also report on investigations of the ferromagnetic properties of the samples leading to the determination of the Curie temperature and coercivity. Finally, they measure the electronic transport properties using Hall measurements.
The paper is clearly written and the experiments are well-described and documented. The thorough experimental characterization is well-supplemented by interpretation of the observed spectra. All in all, the study sheds interesting insights into the properties of Cr-doped BST films, which may be relevant e.g. for the development of magnetic topological insulators.
I would recommend publication of this nice work in Nanomaterials. Below are a few minor points/typos that the authors may want to correct.
-l. 46 "in" in "complementary in electronic properties" should probably be removed.
-l. 200 on Fig. 1f: the Cs P1/2 and Cs P3/2 peaks are rather small. What is the uncertainty in the binding energies extracted from the multi-Lorentz fit?
-the numbers in the legends of Fig. 3 are too small to be readable.
The English is fine.
Author Response
Please read the attachment.

Reviewer 4 Report
Comments and Suggestions for Authors
The research conducted by Pangihutan Gultomand and colleagues is titled "Epitaxial growth and characterization of nanoscale magnetic topological insulators: Cr-doped (Bi0.4Sb0.6)2Te3." This study aligns with the Nanomaterials scope. The comments provide a concise summary of my review,
1. Typographical errors are present in the text at lines 212 and 285.
2. Regarding either the introduction or conclusion sections, the authors have not adequately addressed the detailed application of these structures. In the revised edition of the paper, please include a detailed examination of these aspects, as it would significantly benefit the readers.
Author Response
Please read the attachment.

Round 2
Reviewer 2 Report
Comments and Suggestions for Authors
The revised manuscript is very good for publication.